# Food Consumption in Adolescents and Young Adults: Age-Specific Socio-Economic and Cultural Disparities (Belgian Food Consumption Survey 2014)

**DOI:** 10.3390/nu11071520

**Published:** 2019-07-04

**Authors:** Lucille Desbouys, Karin De Ridder, Manon Rouche, Katia Castetbon

**Affiliations:** 1Research Center in “Epidemiology, Biostatistics and Clinical Trials”, School of Public Health, Université libre de Bruxelles (ULB), 1070 Brussels, Belgium; 2Sciensano, Department of Epidemiology and Public Health and Surveillance, Unit “Lifestyle and chronic diseases”, 1050 Brussels, Belgium

**Keywords:** diet, food, nutrition survey, socio-economic factors, adolescent, young adult

## Abstract

A key issue in nutritional public health policies is to take into account social disparities behind health inequalities. The transition from adolescence toward adulthood is a critical period regarding changes in health behaviors. This study aimed to determine how consumption of four emblematic food groups (two to favor and two to limit) differed according to socio-economic and cultural characteristics of adolescents and young adults living in Belgium. Two non-consecutive 24-h dietary recalls were carried out in a nationally representative sample of 10–39 year old subjects (*n* = 1505) included in the Belgian food consumption survey 2014. Weighted daily mean consumption of “fruits and vegetables”, “whole grain bread and cereals”, “refined starchy food”, and “sugary sweetened beverages” (SSB) was calculated and explored in multivariable linear regressions stratified into four age groups. After adjustment, 10–13 year old adolescents living in less educated households daily consumed lower amounts of “fruits and vegetables” (adjusted mean: 165.6 g/day (95% CI: 125.3–206.0)) and “whole grain bread and cereals” (40.4 g/day (22.9–58.0)), and higher amounts of SSB (309.7 g/day (131.3–488.1) than adolescents of same ages living in more educated households (220.2 g/day (179.8–260.7); 59.0 g/day (40.3–77.8); and 157.8 g/day (1.7–314.0), respectively). The same trends were observed in older groups, along with strong consumption disparities according to region of residency, country of birth, and occupation, with specificities according to age. Our findings suggest the need to better explore such disparities by stage of transition to adulthood, and to adapt nutritional health programs.

## 1. Introduction

Nutrition may play an important role in the increasing burden of cancer, obesity, diabetes, and cardiovascular diseases. In addition to physical activity, a diet rich in vegetables, fruits, and fibers is recommended, along with limited consumption of processed foods high in fat, refined starches and sugars, red and processed meat, sugary sweetened beverages (SSB), and alcohol [1,2]. In Belgium, recent analyses identified the group of 14–17 year old adolescents as having the worst dietary habits when considering adhesion to dietary guidelines [3]. In the 18–39 year old group, conclusions were mitigated: consistent with adolescents, these adults had the highest consumption of nutrient-poor foods (comprising SSB, alcoholic drinks, biscuits and pastries, confectionary and chocolates, salty and fried snacks, etc.) than the youngest and oldest groups, while they also increased consumption of vegetables, fruits, and whole grain bread in comparison to youngest.

Dietary habits are acquired at adolescence and during the transition toward adulthood. Over the past few decades, this key period has increasingly occupied a greater portion of the life course: biological factors related to earlier puberty precede a longer and later appropriation of the adult social role, going beyond the current limit of 19 years [4]. Moreover, the health-related behavior of future adults may be influenced by socio-economic living conditions, family, the school and work environment, and development of a social network [4,5,6,7,8]. To better understand the different factors involved in acquisition of dietary habits, a comparison of adolescent and young adult behavioral determinants is therefore of interest.

In high-income countries, dietary disparities in food group consumption have been widely pointed out in the general population. For instance, whole grains, fresh fruits and vegetables, lean meats, and low-fat dairy products are more likely to be consumed by adults of higher socio-economic status (SES) [9]. As recent literature reviews concluded, food group consumption disparities also involve adolescents. Fruit, vegetable, and dairy intake is higher when parental SES is more favorable [10,11,12,13,14,15], while consumption of SSB and salty or sugary energy-dense products is higher when socio-economic living conditions are lower [10,12,13]. Other disparities according to birthplace, length of time living in a country [16], migration generation [17], and urban/rural place of living [10] have been observed in different countries. However, available information from quality studies on food group consumption disparities in adolescents is scattered, and the variety of studied determinants is limited. Literature focusing on diet disparities in young adults is extremely scarce.

The aim of the present research is to determine how consumption of food groups to favor (fruits and vegetables, whole grain bread, and cereals) and to limit (SSB, refined starchy food) differed according to socio-economic and cultural characteristics of adolescents (according to the common definition, i.e., 10–13 and 14–17 years), young adults (end of adolescence, i.e., 18–25 years), and adults (26–39 years).

## 2. Materials and Methods 

### 2.1. Sampling

We analyzed 10–39 year old subject data from the 2014 Belgian food consumption survey (BFCS), a nationally representative cross-sectional survey conducted in the general population living in Belgium. The BFCS (methods described in detail elsewhere [18]) is part of the “European union (EU) Menu” project coordinated by the European food safety authority (EFSA). To summarize, persons aged 3–64 years were randomly selected from the Belgian national population register, following a multistage stratified sampling procedure. Geographic stratification according to the 11 provinces was used. The number of interviews to be carried out in each province was proportional to the size of each province, and divided into 50 to define the number of municipalities to be selected. Then, within each sampled municipality, individuals were selected by stratifying into 10 age-gender strata following the EFSA age group cut-off recommendations [19]. Data collection was divided equally over the four seasons and seven days of the week so as to integrate seasonal and day-to-day variations in food intake.

### 2.2. Dietary Assessment

A country-specific version of GloboDiet, a computerized 24-h dietary recall (24h-R) program developed and maintained by the international agency for research on cancer, was used [19,20]. Each 24h-R interview was conducted by a trained dietician. The multipass 24h-R method was used with a rapid and consecutive list of consumed foods and recipes at each eating occasion; a description and quantification of such foods and recipes, a summary of the 24h-R, and a description and quantification of dietary supplements. Each recipe was broken down into a list of foods.

All foods were classified according to the FoodEx2 food classification system developed by EFSA [19,21]. For the present analysis, some adaptations were made: for example, the “starchy food” group was divided into “whole grain bread and cereal” and “refined starchy food”, the latter including potatoes and tubers. An SSB group was formed and included all non-alcoholic beverages containing intrinsic sugar, added sugar, flavored milks, and sugary milk substitutes. Mean daily consumption of “fruits and vegetables”, “whole grain bread and cereals”, “refined starchy food”, and SSB was calculated by subject.

Food consumption was linked to the Belgian food composition data (Nubel) and to the Dutch food composition data (NEVO) to estimate total energy intake. The Black method and Goldberg cut-off [22] were used to identify 18–39 year old under-reporters: the mean energy intake of each subject was compared to the basal metabolic rate (BMR), estimated by the Schofield equation [23]. A mean physical activity level of 1.55 was used. Energy intake day-to-day variations, a between-subject BMR variation of 8.5%, and a between-subject physical activity level variation of 15% were considered. Among 10–17 year olds, since the Black method does not take into account energy needs related to growth, under-reporters were those declaring mean energy intake below two standard deviations of the mean energy intake of the sample.

### 2.3. Socio-Economic Status and Cultural Characteristics

Variables related to household type and education level were adapted to adolescents and young adults. Whatever the age, if the subject was still in school during the survey, the number of children in the household included the subject, while the highest education level of the household was the parental education level. If the subject was 18 years old or older and not schooled, the highest education level in the household was defined according to the education level of the subject and his/her partner if applicable. Occupation (of the mother and the 26–39 year old adult subject) was divided into 5 categories, with “inactive” status including students, retired persons, the unemployed, those on sick leave, and the disabled. Working status was grouped into “student”, “active”, and “inactive” categories, as to be adapted to the 18–25 year old group. In addition, household type, country of birth, and main language spoken at home were considered.

### 2.4. Statistical Analyses

Pregnant and breastfeeding women were excluded from analyses; energy under-reporters were included. A weighting factor calculated according to age, gender, day of first dietary recall (weekday or weekend), season, and province of residency, along with sample design, were taken into account in statistical analyses (using the “svyset” function, Stata^®^). All analyses were stratified into four age groups: 10–13 years, 14–17 years, 18–25 years, and 26–39 years. For each food group, mean consumption of the two observation days was calculated. Univariate linear regressions of daily mean consumptions, with SES variables, gender, and region, were systematically adjusted for total energy intake (Appendix A). Those variables with a significance level under 0.20 were included in initial multivariable models. After a manual backward stepwise process, final multivariate linear regressions included variables significantly associated with daily mean consumption (*p*-value <0.05), along with confounding variables (i.e., that is, variables whose removal increased or decreased consumed amounts by more than 10% for other variables included in the model). In addition, once an explanatory variable was retained for an age group, even though it was not significantly associated with the outcome or was not a confounder, this variable (or the one(s) concerning the same topic in other age groups) was also kept for all age groups. Adjusted mean daily consumption was post-estimated using predictive margins, with co-variates being treated as non-fixed [24]. The absence of co-linearity between variables included in multivariate models, normal distribution, homoscedasticity, and linearity of residuals, along with the absence of influence of potential outliers, were graphically verified in unweighted models. All analyses were performed using Stata^®^ version 14 (StataCorp, College Station, TX, USA).

## 3. Results

The total sample under study was composed of 1505 subjects having completed two non-consecutive 24h-R, stratified into 447 10–13, 470 14–17, 233 18–25, and 355 26–39 year olds (Appendix A). One (0.2%) under-reporter was identified among the 10–13 year olds, 3 (0.6%) among the 14–17 year olds, 47 (20.2%) among the 18–25 year olds, and 68 (19.2%) among the 26–39 year olds. Mean total energy intake during the two days of study was 1,783.2 kcal/day (SEM: 28.3), 1979.9 kcal/day (35.7), 2056.2 kcal/day (70.1), and 2033.9 kcal/day (42.4), respectively.

Nearly all adolescents and young adults consumed “fruits and vegetables” and “refined starchy food” at least once during the two days of recall (Table 1). The mean daily consumption of “fruits and vegetables” was statistically higher among 18–25 and 26–39 year olds than among 10–13 year olds. The consumption of “whole grain bread and cereals” and contribution to total starchy food intake were statistically higher among 26–39 year olds than among 10–13 year olds. “Refined starchy food” and SSB consumptions were the highest among 14–17 year olds. “Refined starchy food” contribution to total starchy food intake and SSB contribution to total beverage intake were significantly lower among 26–39 year olds than among 10–13 year olds. 

### 3.1. Fruits and Vegetables

Among 10–13 and 14–17 year olds, and after adjustment, “fruit and vegetable” consumption was lower in households with a secondary or lower education level than households with postgraduate education (Table 2). Regional disparities were observed to the detriment of Wallonia (in 10–13 year olds) and the Brussels-Capital region (in 14–17 year olds) inhabitants, which consumed lower amounts of “fruits and vegetables” than the Flemish. In the 14–17 year old group only, boys, adolescents whose mothers were manual workers and those born in Belgium consumed significantly fewer “fruits and vegetables” daily than girls, subjects whose mothers had a managerial or academic occupation and those born in EU and outside the EU, respectively. Finally, among 26–39 year olds, subjects living in a single-parent family and those born in Belgium had lower “fruit and vegetable” consumption than those couples without children and born outside the EU, respectively.

### 3.2. Whole Grain Bread and Cereals

After adjustment, the region of residency was associated with daily mean consumption of “whole grain bread and cereals”, with Wallonia (in all age groups) and Brussels-Capital region (in 10–13 and 14–17 year old groups) residents consuming smaller amounts than the Flemish (Table 3). Among 10–13 and 14–17 year olds, “whole grain bread and cereal” consumption was lower in households with secondary or lower education levels than households with postgraduate education. Among 14–17 and 18–25 year olds, subjects born in Belgium ate significantly fewer “whole grain bread and cereals” than those born elsewhere in the EU. In the 14–17 year old group only, adolescents whose mothers had a managerial or academic occupation consumed significantly fewer “whole grain bread and cereals” daily than subjects whose mothers were manual workers, employees, or with an intermediate occupation.

### 3.3. Refined Starchy Food

After adjustment, in all age groups, except in the 18–25 year old group, subjects living in Wallonia consumed higher amounts of “refined starchy food” than the Flemish (Table 4). Males consumed generally higher amounts of “refined starchy food” than females, significantly only among 14–17 and 26–39 year olds. In the 18–25 year old group, subjects speaking mixed languages, including French or Dutch, consumed higher amounts of “refined starchy food” than those speaking exclusively French and/or Dutch. 

### 3.4. Sugary Sweetened Beverages

In all age groups and after adjustment, households or responders with a secondary education level or less consumed higher amounts of SSB daily than those with a postgraduate education, or a bachelor’s degree depending on the age group (Table 5). Indeed, education level gradients were observed in the 10–13 and 14–17 year old groups. Among 14–17 and 18–25 year olds, subjects born in Belgium significantly consumed higher amounts of SSB than those born outside the EU. Among 18–25 year olds, subjects living in Flanders significantly consumed higher amounts of SSB than those living in the Brussels-Capital region. Among 26–39 year olds, inactive and manual workers showed higher consumption of SSB than employees or those with an intermediate occupation.

## 4. Discussion

Our aim was to determine differences in consumption of four food groups according to socio-economic and cultural characteristics of adolescents and young adults living in Belgium. In a representative sample in which diet was measured with two non-consecutive 24h-R, consumption of food groups to favor (fruits and vegetables, whole-grain products) increased with age. Moreover, consumption of food groups to limit (refined starchy food and SSB) was the highest among older adolescents (14–17 years), then decreased with adult age. As in other high-income countries, diet disparities in fruit, vegetable, and SSB consumption were observed, to the detriment of less well-educated subjects. In addition, our study provides new findings on whole grain product consumption disparities in all age groups. Strong regional disparities were found, independently of SES and for all food groups. Furthermore, our results indicate that oldest adolescents and young adults who were born in Belgium had less favorable consumption than those born abroad, either within or outside of the EU. Overall, the socio-economic and cultural influences upon food group consumption differ according to age group.

Overall, findings related to education and occupation are consistent with disparities observed in other high-income countries [10,11,12,13,15]. Education is considered to reflect health and nutrition literacy, i.e., the ability to appropriate nutritional information and to implement behavior accordingly [6]. Occupation is associated with the potential influence upon dietary behavior of the work environment, conditions, and the social network, along with social standing [25]. In addition, education is a determinant of occupation (and income, unavailable in our study); thus, all these indicators are interrelated, but are independently involved in dietary disparities [9,26,27]. In all age groups, and particularly among adolescents in the studied sample, less-well-educated households and subjects ate smaller amounts of healthy products and higher amounts of unhealthy products than the more educated. These results are consistent with recent studies among adolescents [10,11,12,13] and young adults [28]. However, disparities in whole grain product consumption had rarely been studied previously: one German study reported no significant association with parental education [29]. In line with previous studies [9,11,12,26], occupation was involved in certain dietary disparities, but not in all age or food groups. Manual workers and inactive subjects, and older adolescents with such parents, were more likely to have an unhealthy diet (higher amounts of SSB and smaller amounts of fruits and vegetables, respectively) than other occupational and active categories. However, older adolescents whose mothers had the highest occupational status were those least consuming whole grain products, which would require further investigations in order to be explained.

Other new insights have emerged from our findings. Wide dietary disparities were encountered according to the region of residency in all age groups and independently of socio-economic conditions. Flanders is socio-economically more advantaged in comparison to Wallonia and the Brussels-Capital region in terms, for instance, of unemployment rate, poverty, and social exclusion [30]. Walloon inhabitants (mainly French-speaking) generally had a less healthy diet than the Flemish (mainly Dutch speakers), which had been previously shown [31]. In multilingual, multiregional Switzerland, substantial differences in diet were found according to linguistic region: indeed, in the 18–75 year old population, weighted daily mean intake of vegetables was significantly higher in German and French regions than in the Italian region, but the daily mean intake of “soft drinks” was higher in the German region than in the French and Italian regions [32]. Authors pointed out the influence of dietary habits from bordering countries: the diet observed in each Swiss region was comparable to that in the neighboring country. Indeed, a parallel could be made between the German community in Switzerland and the Belgian Flanders community, since they consumed higher amounts of vegetables and SSB daily than their French-speaking counterparts. In another study on the diversity of dietary patterns in European countries, the French population consumed fewer soft drinks than other Europeans; in the Netherlands, juice and soft drink consumption was higher than the European mean [20]. Nevertheless, in the Netherlands, vegetable and fruit consumption was lower than in other European countries [33], contradicting the hypothesis of cultural influence from bordering countries on Belgian dietary habits. Regional dietary specificities in Belgium may therefore be only partly explained by neighboring influences, possibly combined with various changes in regional public health policies.

Furthermore, being born in Belgium, as opposed to the EU or outside the EU (depending on the food group), was globally associated with a less healthy diet (lower amounts of fruits, vegetables, and whole grain products, and higher amounts of SSB), mainly among 14–17 and 18–25 year olds. Previous studies on migration disparities in diet showed that migrants—and especially recent migrants [16]—had higher dietary quality scores [34], healthier patterns [35], and consumed more vegetables than natives or less recent migrants [16]. In one study, foreign-born subjects also ate more SSB than natives [16], while this was not the case in the present study. Here, we only made a distinction between migrants from EU and outside the EU, but more detailed information on country of birth and age at arrival in the host country should also be investigated, so as to better explore potential acculturation phenomena [36,37,38] in a multicultural country such as Belgium.

We also sought to determine whether dietary disparities were life-stage-specific. We observed that consumption of four emblematic food groups improved with age, being the less favorable among 14–17 year olds. Moreover, in such group of older adolescents, the socio-economic and cultural characteristics of diet disparities were the most diverse. However, complex interpretation of findings in 18–25 year olds and limited sample size made it difficult to identify age specificities. It would be useful to study more in-depth factors involved in dietary behavior during this rather lengthy stage of “semi-dependency” in young adults, i.e., the influence of family transmission, school, or work environment, and individual health and well-being. In addition, changes in diet occurring when the subject becomes responsible for others (partner, children) would be of interest.

For a relevant interpretation of our findings, some limitations should be noted. By definition, collected SES variables differed according to age group, with certain variables specific to life stage. For example, parental occupation was collected in 10–13 and 14–17 year old groups, while that of the subject themselves was collected in adult groups. In the 18–25 year old group, occupational categories such as managerial or academic were not plausible, so the working status was therefore coded into “student”, “inactive”, and “active”. The interpretation of occupational disparities between age groups was therefore limited even if some common trends were observed in the hierarchy of status. In addition, the 18–25 year old group was composed of two-thirds of students living with their parents or dependent on their family (median age = 20.3 years), and one-third of non-students living independently (median age = 23.3 years). This heterogeneity also limited interpretation of potential disparities between food consumption and household type in this age group. Additional stratified analyses according to student status would have been useful, but were not feasible due to the small number of subjects concerned.

Language mainly spoken at home was only associated with refined starchy food consumption in this study. Categorization of this cultural variable aimed to indirectly and partially explore differences in literacy and its potential influence on diet [39]. However, the main language spoken in the household was asked in a semi-open question (Dutch or French option, since they are the two main languages among the three official languages in Belgium according to region of residency vs. another language, with open field to specify). Numerous subjects indicated that they spoke more than one language, without specification of hierarchy; lack of accuracy may therefore be suspected.

Finally, based on two non-consecutive recall days, virtually the entire adolescent and young adult population under study consumed refined starchy food and fruits and vegetables, while up to three-fourths of the sample consumed SSB, and up to two-thirds ate whole grain bread and cereals. In terms of starchy food consumption, the challenge lies in convincing the entire population to more often replace refined product consumption with whole grain products, since overall consumption is low. For fruits, vegetables, and SSB, the wide socio-economic and cultural disparities in all age groups suggest that accessibility [9] and affordability [40] of such products, along with associated perception of availability [41] and benefits to health [42], are factors that must be considered. Less well-educated adolescents and young adults born in Belgium identified as currently consuming fewer fruits, vegetables, and whole grain products, and more SSB, should be specifically targeted.

## 5. Conclusions

The present study emphasizes socio-economic and cultural disparities in the consumption of four food groups in adolescents and young adults living in Belgium. A healthier diet pattern was observed with age, and our findings suggest that certain disparities may be life-stage-specific. Further analyses addressing other food group consumption (such as for instance meat, fish and eggs, or dairy products) or using a prospective design are needed to better understand changes in dietary behavior occurring between adolescence and young adulthood. Overall, a lower education level, birth in Belgium, and living in Wallonia (excepted for SSB consumption) were independently associated with less healthy dietary habits in all age groups. These characteristics, which had not been previously elucidated, along with regional specificities, should be taken into account in future public nutrition interventions.

## Figures and Tables

**Table 1 nutrients-11-01520-t001:** Consumption, at least once during the two days of recall, mean daily consumption, and contribution to total intake of four food groups according to age group. Belgian food consumption survey 2014.

Food Groups	Age Category				
	10–13 years *n* = 447	14–17 years *n* = 470	18–25 years *n* = 233	26–39 years *n* = 355	*p* ^a^
**Fruits and vegetables**					
Consumption at least once during the 2 days (%)	97.8	98.1	98.5	99.5	
Mean daily consumption (g/day (SEM))	190.2 (6.8) *	196.7 (7.6)	**221.1 (12.1)**	**269.6 (11.5)**	**<0.001**
**Whole grain bread and cereals**					
Consumption at least once during the 2 days (%)	51.6	52.0	55.9	62.5	
Mean daily consumption (g/day (SEM))	37.3 (3.2) *	42.0 (3.3)	43.1 (4.9)	**63.5 (4.9)**	**<0.001**
Mean contribution to total starchy food intake (% (SEM))	16.6 (1.2) *	15.5 (1.1)	16.9 (1.8)	**24.0 (1.6)**	**<0.001**
**Refined starchy food**					
Consumption at least once during the 2 days (%)	99.4	100.0	99.8	99.2	
Mean daily consumption (g/day (SEM))	188.4 (4.9) *	**226.6 (6.0)**	**220.7 (10.9)**	203.4 (7.4)	**<0.001**
Mean contribution to total starchy food intake (% (SEM))	83.4 (1.2) *	84.5 (1.1)	83.1 (1.8)	**76.0 (1.6)**	**<0.001**
**Sugary sweetened beverages**					
Consumption at least once during the 2 days (%)	71.9	74.4	74.1	60.3	
Mean daily consumption (g/day (SEM))	258.7 (17.4) *	**327.8 (17.5)**	322.4 (33.3)	240.3 (21.1)	**<0.01**
Mean contribution to total beverage intake (% (SEM))	24.3 (21.4) *	26.9 (1.3)	20.2 (1.9)	**14.1 (1.1)**	**<0.001**

^a^ Test of difference in means compared with the reference group; * Reference group; bold: category for which consumption statistically significantly differed from the reference category (*p* < 0.05).

**Table 2 nutrients-11-01520-t002:** Adjusted ^a^ mean consumption (g/day) of fruits and vegetables according to socio-economic, cultural characteristics, and age group. Belgian food consumption survey 2014.

Characteristics	10–13 years, *n* = 435		14–17 years, *n* = 460		18–25 years, *n* = 229		26–39 years, *n* = 352	
	Mean (95%CI) ^b^	*p*	Mean (95%CI) ^c^	*p*	Mean (95%CI) ^d^	*p*	Mean (95%CI) ^e^	*p*
**Gender**		0.80		**<0.01**		0.34		0.39
**Male**	183.7 (145.3–222.0)		**187.8 (137.5–238.2)**		211.6 (179.9–243.4)		244.5 (205.2–283.8)	
**Female**	187.0 (149.3–224.7) *		234.1 (184.0–284.1) *		234.6 (198.9–270.4) *		266.6 (238.3–294.9) *	
**Household type**								
**Two–parent family**	185.1 (146.4–223.8) *	0.93	213.9 (164.9–262.9) *	0.41	–		-	
**Single–parent family**	186.4 (154.0–218.8)		201.4 (153.3–249.6)					
**Single**					184.2 (91.6–276.8)	0.37	195.7 (140.1–251.2)	**<0.01**
**Single–parent family**					185.4 (130.9–239.9)		**196.8 (142.5–251.2)**	
**Couple without children**	-		-		265.4 (190.6–340.2)		281.9 (219.6–344.1) *	
**Two–parent family**					227.5 (189.5–265.5) *		263.0 (235.8–290.1)	
**Other**					202.9 (135.9–270.0)		329.6 (230.7–428.5)	
**Highest education level in the household**		**<0.01**		**0.02**		0.07		
**2dary education or lower**	**165.6 (125.3–206.0)**		**184.6 (135.8–233.4)**		**182.9 (141.7–224.2)**			
**Bachelor’s degree or equivalent**	**169.2 (128.1–210.3)**		223.8 (172.0–275.6)		231.8 (189.8–273.8)		-	
**Postgraduate education**	220.2 (179.8–260.7) *		224.9 (172.1–277.7) *		256.9 (207.9–305.9) *			
**Education level of the responder**								0.27
**2dary education or lower**							236.4 (199.3–273.5)	
**Bachelor’s degree or equivalent**	-		-		-		259.8 (221.8–297.9)	
**Postgraduate education**							283.0 (241.4–324.7) *	
**Maternal occupation**		0.54		**<0.01**				
**Inactive**	182.1 (151.6–212.6)		185.4 (159.7–211.2)					
**Manual worker**	178.4 (137.0–219.8)		**136.7 (107.2–166.2)**					
**Self–employed**	193.6 (156.7–230.4)		205.4 (164.9 –245.8)		-		-	
**Employee or intermediate**	209.1 (188.7–229.4)		204.4 (184.4–224.4)					
**Managerial or academic**	198.7 (149.2–248.2) *		243.6 (146.2–341.0) *					
**No mother declared**	182.5 (139.6–225.4)		214.8 (157.6–272.1)					
**Working status**						0.68		
**Student**					216.7 (184.6–248.8)			
**Inactive**	-		-		202.9 (133.9–272.0)		-	
**Active**					234.3 (195.8–272.9) *			
**Occupation**								0.24
**Inactive**							228.1 (180.2–276.0)	
**Manual worker**							228.9 (177.5–280.3)	
**Self–employed**	-		-		-		271.0 (211.2–330.8)	
**Employee or intermediate**							286.8 (252.3–321.3)	
**Managerial or academic**							260.6 (195.8–325.4) *	
**Country of birth**		0.28		**<0.001**		0.69		**0.02**
**Belgium**	184.6 (148.6–220.6) *		203.3 (155.7–250.9) *		222.9 (197.4–248.4) *		246.6 (220.6–272.7) *	
**EU**	174.5 (121.7–227.3)		**255.5 (190.0–320.9)**		200.6 (145.5–255.8)		281.7 (197.4–366.0)	
**Outside the EU**	215.4 (159.4–271.4)		**350.6 (257.0–444.2)**		257.0 (127.3–386.8)		**389.7 (293.8–485.6)**	
**Language spoken at home**		0.97		0.32		0.18		0.21
**French and/or Dutch**	185.8 (149.7–221.9) *		207.5 (160.1–255.0) *		217.9 (193.2–242.5) *		261.8 (236.8–286.9) *	
**Mixed incl. French or Dutch**	182.3 (136.2–228.4)		238.1 (169.9–306.3)		297.1 (217.3–377.0)		198.0 (94.7–301.2)	
**Language other than French or Dutch**	180.2 (90.5–270.0)		260.0 (158.7–361.3)		238.5 (84.3–392.8)		166.3 (60.5–272.1)	
**Region of residency**		**<0.001**		**0.04**		0.14		0.09
**Flanders**	202.6 (164.3–240.9) *		220.1 (170.2–270.1) *		235.6 (204.9–266.4) *		270.5 (242.1–298.8) *	
**Brussels**	191.9 (136.8–247.0)		**170.1 (114.1–226.0)**		237.4 (129.2–345.6)		245.2 (122.6–367.8)	
**Wallonia**	**148.1 (110.5–185.6)**		205.2 (156.2–254.2)		193.1 (155.7–230.4)		**227.5 (197.4–257.6)**	

^a^ Adjusted for total energy intake and other variables included in the model; ^b^ R^2^ = 13.6%; ^c^ R^2^ = 17.3%; ^d^ R^2^ = 12.1%; ^e^ R^2^ = 11.9%; * reference category; bold: category for which consumption statistically significantly differed from reference category (*p* < 0.05); -: variable not concerning the age group.

**Table 3 nutrients-11-01520-t003:** Adjusted ^a^ mean consumption (g/day) of whole grain bread and cereals according to socio-economic and cultural characteristics, and age group. Belgian food consumption survey 2014.

Characteristics	10–13 years, *n* = 435		14–17 years, *n* = 463		18–25 years, *n* = 229		26–39 years, *n* = 352	
	Mean (95%CI) ^b^	*p*	Mean (95%CI) ^c^	*p*	Mean (95%CI) ^d^	*p*	Mean (95%CI) ^e^	*p*
**Highest education level in the household**		**0.02**		**0.03**		0.16		
2dary education or lower	**40.4 (22.9–58.0)**		**46.6 (25.7–67.5)**		32.5 (21.4–43.6)			
Bachelor’s degree or equivalent	49.2 (32.0–66.4)		57.1 (35.6–78.5)		42.1 (27.8–56.5)		-	
Postgraduate education	59.0 (40.3–77.8) *		64.8 (43.7–85.9) *		53.3 (32.9–73.6) *			
**Education level of the responder**								0.77
2dary education or lower							53.5 (35.0–72.1)	
Bachelor’s degree or equivalent	-		-		-		61.0 (45.1–76.8)	
Postgraduate education							65.2 (45.2–85.2) *	
**Maternal occupation**		0.12		**0.02**	-		-	
Inactive	39.6 (25.7–53.6)		35.9 (24.2–47.5)					
Manual worker	28.8 (19.3–38.3)		**39.6 (27.2–52.0)**					
Self–employed	54.3 (28.4–80.1)		31.7 (16.3–47.2)					
Employee or intermediate	38.4 (31.4–45.4)		**44.7 (36.2–53.2)**					
Managerial or academic	25.4 (7.4–43.5) *		17.7 (2.0–33.5) *					
No mother declared	50.9 (32.5–69.4)		**58.2 (36.1–80.3)**					
**Working status**						0.65		
Student					38.6 (28.9–48.4)			
Inactive	-		-		35.7 (10.4–61.1)		-	
Active					47.2 (29.6–64.7) *			
**Occupation**								0.25
Inactive							48.2 (29.9–66.5)	
Manual worker							65.1 (35.2–95.0)	
Self–employed	-		-		-		38.8 (9.4–68.1)	
Employee or interm.							68.3 (53.0–83.6) *	
Managerial or academic							61.2 (34.9–87.6)	
**Country of birth**		0.30		**0.02**		**0.02**		0.86
Belgium	47.5 (31.5–63.5) *		52.2 (33.4–71.1) *		39.3 (28.6–50.1) *		59.8 (50.4–69.2) *	
EU	69.4 (26.1–112.8)		**94.1 (56.2–131.9)**		**104.1 (60.0–148.2)**		52.8 (17.5–88.2)	
Outside the EU	62.3 (31.2–93.4)		84.7 (33.1–136.3)		31.6 (11.6–51.7)		50.6 (10.7–90.5)	
**Region of residency**		**<0.001**		**<0.001**		**<0.001**		**<0.001**
Flanders	60.1 (42.9–77.4) *		66.0 (45.8–86.3) *		55.3 (43.5–67.0) *		72.0 (59.2–84.7) *	
Brussels	**39.8 (20.5–59.2)**		**40.9 (17.3–64.6)**		35.2 (11.2–59.2)		50.6 (18.5–82.6)	
Wallonia	**30.7 (13.2–48.2)**		**40.6 (19.2–61.9)**		**18.6 (3.0–34.2)**		**33.6 (20.9–46.3)**	

^a^ Adjusted for total energy intake and other variables included in the model; ^b^ two influent outliers excluded, R^2^ = 18.1%; ^c^ R^2^ = 16.3%; ^d^ R^2^ = 22.8%; ^e^ R^2^ = 12.0%; * reference category; bold: category for which consumption statistically significantly differed from reference category (*p* < 0.05); -: variable not concerning the age group.

**Table 4 nutrients-11-01520-t004:** Adjusted ^a^ mean consumption (g/day) of refined starchy food according to socio-economic and cultural characteristics, and age group. Belgian food consumption survey 2014.

Characteristics	10–13 years, *n* = 447		14–17 years, *n* = 470		18–25 years, *n* = 233		26–39 years, *n* = 355	
	Mean (95%CI) ^b^	*p*	Mean (95%CI) ^c^	*p*	Mean (95%CI) ^d^	*p*	Mean (95%CI) ^e^	*p*
**Gender**		0.77		**<0.001**		0.08		**0.04**
Male	198.1 (185.2–211.0)		**239.0 (223.9–254.1)**		225.2 (195.5–254.8)		**208.6 (186.9–230.2)**	
Female	195.6 (182.7–208.5) *		198.7 (185.5–211.8) *		193.0 (176.4–209.7) *		180.1 (165.2–194.9) *	
**Language spoken at home**		0.13		0.09		**0.02**		0.45
French and/or Dutch	194.0 (183.9–204.1) *		219.3 (208.5–230.2) *		205.3 (189.0–221.5) *		196.8 (183.2–210.3) *	
Mixed incl. French or Dutch	209.1 (182.1–236.1)		234.0 (201.5–266.4)		**271.1 (225.8–316.4)**		188.6 (142.5–234.7)	
Language other than French or Dutch	248.3 (192.1–304.6)		**175.8 (133.9–217.8)**		195.5 (106.6–284.4)		138.8 (51.4–226.1)	
**Region of residency**		**<0.001**		**<0.01**		0.80		**0.04**
Flanders	183.5 (171.5–195.5) *		204.3 (192.1–216.4) *		205.2 (180.8–229.6) *		182.8 (168.0–197.6) *	
Brussels	212.8 (183.3–242.2)		243.0 (201.0–284.9)		209.7 (173.4–246.0)		189.6 (134.8–244.4)	
Wallonia	**218.4 (204.1–233.8)**		**239.5 (221.8–257.2)**		216.4 (196.4–236.5)		**219.0 (194.4–243.4)**	

^a^ Adjusted for total energy intake and other variables included in the model; ^b^ R^2^ = 23.1%; ^c^ R^2^ = 29.9%; ^d^ R^2^ = 40.1%; ^e^ R^2^ = 20.9%; * reference category; bold: category for which consumption statistically significantly differed from reference category (*p* < 0.05); -: variable not concerning the age group.

**Table 5 nutrients-11-01520-t005:** Adjusted ^a^ mean consumption (g/day) of sugary sweetened beverages according to socio-economic and cultural characteristics, and age group. Belgian food consumption survey 2014.

Characteristics	10–13 years, *n* = 435		14–17 years, *n* = 460		18–25 years, *n* = 229		26–39 years, *n* = 352	
	Mean (95%CI) ^b^	*p*	Mean (95%CI) ^c^	*p*	Mean (95%CI) ^d^	*p*	Mean (95%CI) ^e^	*p*
**Household type**								
Two–parent family	234.5 (72.0–396.9) *	0.33	360.9 (252.4–469.4) *	0.92	-		-	
Single–parent family	270.3 (109.0–431.5)		356.2 (263.6–448.9)					
Single					585.6 (194.1–977.1)	0.26	288.9 (196.6–381.2)	0.35
Single–parent family					382.6 (220.0–545.1)		162.8 (53.2–272.5) *	
Couple without children	-		-		267.0 (58.0–476.1)		307.6 (223.1–392.1)	
Two–parent family					250.9 (178.2–323.6) *		260.2 (186.5–334.0)	
Other					417.5 (201.2–633.7)		286.8 (140.8–432.8)	
**Highest education level in the household**		**<0.001**		**<0.01**		**0.02**		
2dary education or lower	**309.7 (131.3–488.1)**		**403.9 (283.7–524.2)**		**398.1 (291.9–504.2)**			
Bachelor’s degree or equivalent	**260.6 (101.3–420.0)**		**402.8 (292.2–513.4)**		225.8 (141.3–310.3) *		-	
Postgraduate education	157.8 (1.7–314.0) *		275.5 (167.5–383.4) *		300.5 (181.6–419.5)			
**Education level of the responder**								**0.03**
2dary education or lower							**336.6 (241.2–432.0)**	
Bachelor’s degree or equivalent	-		-		-		221.7 (167.8–275.7)	
Postgraduate education							195.6 (139.2–251.9) *	
**Maternal occupation**		0.33		0.46				
Inactive	263.1 (206.9–319.3)		356.9 (282.9–431.0)					
Manual worker	338.0 (245.7–430.3)		405.8 (255.4 –556.2)					
Self–employed	205.6 (121.3–290.0)		320.3 (213.8–426.7)					
Employee or intermediate	289.2 (242.7–335.6)		291.4 (245.5–337.3)		-		-	
Managerial or academic	242.5 (133.6–351.3) *		251.9 (117.8–386.1) *					
No mother declared	233.5 (40.2–426.7)		369.2 (248.8–489.7)					
**Working status**						0.76		
Student					287.6 (200.5–374.8)			
Inactive	-		-		391.0 (134.0–647.9)		-	
Active					295.1 (200.8–389.4) *			
**Occupation**								**0.01**
Inactive							**345.5 (209.0–481.9)**	
Manual worker							**311.6 (200.3–422.9)**	
Self–employed	-		-		-		289.5 (167.0–412.0)	
Employee or intermediate							181.8 (139.8–223.7) *	
Managerial or academic							215.3 (119.3–311.2)	
**Country of birth**		0.27		**0.02**		**<0.01**		0.62
Belgium	242.2 (83.1–401.3) *		369.7 (270.0–469.5) *		318.5 (245.7–391.3) *		257.2 (201.2–313.2) *	
EU	187.6 (−1.8–377.0)		267.2 (121.6–412.8)		327.2 (38.4–616.0)		314.6 (179.6–449.6)	
Outside the EU	292.9 (97.2–488.5)		**237.0 (98.8–375.1)**		**85.0 (**−**73.6–243.5)**		334.2 (114.7–553.6)	
**Region of residency**		0.08		0.23		**<0.01**		0.05
Flanders	248.9 (91.4–406.5) *		381.7 (279.9–483.5) *		346.1 (256.3–435.9) *		265.7 (216.2–315.2)	
Brussels	**167.5 (11.1–324.0)**		332.5 (203.4–461.5)		**111.0 (–13.8–235.8)**		**175.0 (81.0–269.1)**	
Wallonia	247.9 (71.9–424.0)		323.0 (210.8–435.3)		297.5 (197.2–397.8)		291.6 (199.6–383.7) *	

^a^ Adjusted for total energy intake and other variables included in the model; ^b^ R^2^ = 21.7%; ^c^ R^2^ = 14.6%; ^d^ R^2^ = 24.7%; ^e^ R^2^ = 26.2%; * reference category; bold: category for which consumption statistically significantly differed from reference category (*p* < 0.05); -: variable not concerning the age group.

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
