# Peer review of "Food Consumption in Adolescents and Young Adults: Age-Specific Socio-Economic and Cultural Disparities (Belgian Food Consumption Survey 2014)"

_nutrients, 2019, doi:10.3390/nu11071520_

Round 1

Reviewer 1 Report

The study aimed to determine how consumption of four chosen food groups differs according to socio-economic and cultural characteristics in adolescents and young adults living in Belgium.

The idea of the topic is interesting, however, from the nutritional point of view the use of the terms "healthy/unhealthy food” is incorrect.  Food consists of many components that can have a positive and/or negative effect on the human body, it may depend on health status, treatments etc.

Therefore, in the title, aim and throughout the manuscript please remove above mention terms and instead of these, you may use "chosen/selected food groups".

Could you explain the reason to select for the study of these four food groups but no other, for example, meat, fish, dairy products, sweets etc.?

Author Response

Dear Editor and Reviewers,

We would like to thank you for your commitment to improve our manuscript. We have taken into account your comments and have adapted the manuscript accordingly (changes are highlighted in yellow). Please find below our answers to each comment.

In addition, we have detected an error in the calculation of the proportions of under-reporters. We have corrected the manuscript.

We hope that our revised version will fulfill the requirements of Nutrients.

Sincerely Yours,

Lucille Desbouys, the Corresponding Author.

_________________________________________________________________________________

Response to Reviewer 1 comments

Point 1: The study aimed to determine how consumption of four chosen food groups differs according to socio-economic and cultural characteristics in adolescents and young adults living in Belgium.

The idea of the topic is interesting, however, from the nutritional point of view the use of the terms "healthy/unhealthy food” is incorrect.  Food consists of many components that can have a positive and/or negative effect on the human body, it may depend on health status, treatments etc.

Therefore, in the title, aim and throughout the manuscript please remove above mention terms and instead of these, you may use "chosen/selected food groups".

Response 1: Thank you for your comments. We agree that these terms may be discussable because they summarize too simply what we aimed to address. We have removed "healthy/unhealthy food groups" from the title and the abstract, and we have replaced these with "food groups to favor/to limit" in the abstract, objective (lines 63-64) and at the beginning of the discussion (lines 204-205).

Point 2: Could you explain the reason to select for the study of these four food groups but no other, for example, meat, fish, dairy products, sweets etc.?

Response 2: The four food groups were chosen based on the following reasons: the recommendations to consume enough fruit and vegetables, to promote whole grain product consumption and to limit SSB consumption have been internationally established based on the available evidence. The choice to also study the consumption of refined starchy food was made in order to make a comparison with whole grain products. Indeed, consumption occasions of these two groups are close; we wanted to investigate whether their socioeconomic and cultural correlates could be different since they can be substituted one for the other.

Concerning other food groups, for instance meat/fish/eggs, dairy products or energy-dense food, the interpretation of these is more complex: relationships with health is less established, and we should have had analyzed subgroups (since recommendations may be more specific). In addition, we have chosen to not overload the paper. We have added a sentence about this topic in Conclusions (lines 319-320).

Reviewer 2 Report

This manuscript is very interesting because it describes eating habits in young people. But I am confused. In abstract authors describe research sample (10-39 years) but in the part 2.1 Sampling they describe "persons aged 3 to 64". I think that they should be explain why they choosed a part of sample.

Limitations are part of discussion. It can be confuse for readers. I recommend the limitations explain in independently on discussion.

Author Response

Dear Editor and Reviewers,

We would like to thank you for your commitment to improve our manuscript. We have taken into account your comments and have adapted the manuscript accordingly (changes are highlighted in yellow). Please find below our answers to each comment.

In addition, we have detected an error in the calculation of the proportions of under-reporters. We have corrected the manuscript.

We hope that our revised version will fulfill the requirements of Nutrients.

Sincerely Yours,

Lucille Desbouys, the Corresponding Author.

_________________________________________________________________________________

Response to Reviewer 2 comments

Point 1: This manuscript is very interesting because it describes eating habits in young people. But I am confused. In abstract authors describe research sample (10-39 years) but in the part 2.1 Sampling they describe "persons aged 3 to 64". I think that they should be explain why they choosed a part of sample.

Response 1: Thank you for your comments. In fact, the sample mentioned in section 2.1 is the overall sample from which is extracted our study sample of adolescents and young adults as the objective. To clarify, we have added “10-to-39-year-old subjects” at the beginning of the Materials and Methods section (line 70).

Point 2: Limitations are part of discussion. It can be confuse for readers. I recommend the limitations explain in independently on discussion.

Response 2: The paragraph corresponding to the limitations has been moved to the end of the Discussion (lines 285-304). Some limitations are discussed in two other paragraphs (lines 272-275 and 279-280) but for the clarity of the reasoning, we decided to keep them at the same place.
